# Model-based Trajectory Stitching for Improved Offline Reinforcement Learning

**Charles A. Hepburn**[1]   **Giovanni Montana**[1,2]
[1]University of Warwick   [2]Alan Turing Institute
{Charlie.Hepburn,g.montana}@warwick.ac.uk

## Abstract

In many real-world applications, collecting large and high-quality datasets may be too costly or impractical. Offline reinforcement learning (RL) aims to infer an optimal decision-making policy from a fixed set of data. Getting the most information from historical data is then vital for good performance once the policy is deployed. We propose a model-based data augmentation strategy, Trajectory Stitching (TS), to improve the quality of sub-optimal historical trajectories. TS introduces unseen actions joining previously disconnected states: using a probabilistic notion of state reachability, it effectively 'stitches' together parts of the historical demonstrations to generate new, higher quality ones. A stitching event consists of a transition between a pair of observed states through a synthetic and highly probable action. New actions are introduced only when they are expected to be beneficial, according to an estimated state-value function. We show that using this data augmentation strategy jointly with behavioural cloning (BC) leads to improvements over the behaviour-cloned policy from the original dataset. Improving over the BC policy could then be used as a launchpad for online RL through planning and demonstration-guided RL.

## 1 Introduction

Behavioural cloning (BC) [51, 52] is one of the simplest imitation learning methods to obtain a decision-making policy from expert demonstrations. BC treats the imitation learning problem as a supervised learning one. Given expert trajectories - the expert's paths through the state space - a policy network is trained to reproduce the expert behaviour: for a given observation, the action taken by the policy must closely approximate the one taken by the expert. Although a simple method, BC has shown to be very effective across many application domains [51, 55, 32, 50], and has been particularly successful in cases where the dataset is large and has wide coverage [13]. An appealing aspect of BC is that it is applied in an offline setting, using only the historical data. Unlike reinforcement learning (RL) methods, BC does not require further interactions with the environment. Offline policy learning can be advantageous in many circumstances, especially when collecting new data through interactions is expensive, time-consuming or dangerous; or in cases where deploying a partially trained, sub-optimal policy in the real-world may be unethical, e.g. in autonomous driving and medical applications.

BC extracts the behaviour policy which created the dataset. Consequently, when applied to sub-optimal data (i.e. when some or all trajectories have been generated by non-expert demonstrators), the resulting behavioural policy is also expected to be sub-optimal. This is due to the fact that BC has no mechanism to infer the importance of each state-action pair. Other drawbacks of BC are its tendency to overfit when given a small number of demonstrations and the state distributional shift between training and test distributions [54, 13]. In the area of imitation learning, significant efforts have been made to overcome such limitations, however the available methodologies generally rely

3rd Offline Reinforcement Learning Workshop at Neural Information Processing Systems, 2022.

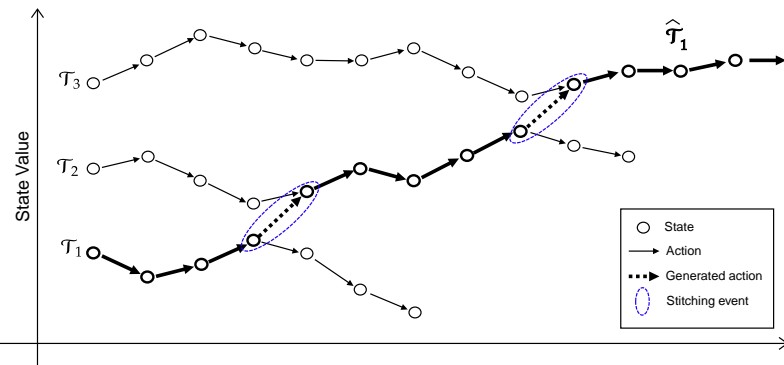

Figure 1: Simplified illustration of Trajectory Stitching. Each original trajectory (a sequence of states and actions) in the dataset $\mathcal{D}$ is indicated as $\mathcal{T}_i$ with $i = 1, 2, 3$. A first stitching event is seen in trajectory $\mathcal{T}_1$ whereby a transition to a state originally visited in $\mathcal{T}_2$ takes place. A second stitching event involves a jump to a state originally visited in $\mathcal{T}_3$. At each event, jumping to a new state increases the current trajectory's future expected returns. The resulting trajectory (in bold) consists of a sequence of states, all originally visited in $\mathcal{D}$, but connected by imagined actions; it replaces $\mathcal{T}_1$ in the new dataset.

on interacting with the environment [54, 18, 28, 47]. So, a question arises: can we help BC infer a superior policy only from available sub-optimal data without the need to collect additional expert demonstrations?

Our investigation is related to the emerging body of work on offline RL, which is motivated by the aim of inferring expert policies with only a fixed set of sub-optimal data [46, 48]. A major obstacle towards this aim is posed by the notion of *action distributional shift* [23, 43, 48]. This is introduced when the policy being optimised deviates from the behaviour policy, and is caused by the action-value function overestimating out-of-distribution (OOD) actions. A number of existing methods address the issue by constraining the actions that can be taken. In some cases, this is achieved by constraining the policy to actions close to those in the dataset [23, 43, 60, 31, 67, 21], or by manipulating the action-value function to penalise OOD actions [45, 1, 39, 62]. In situations where the data is sub-optimal, offline RL has been shown to recover a superior policy to BC [23, 44]. Improving BC will in turn improve many offline RL policies that rely on an explicit behaviour policy of the dataset [2, 64, 21].

In contrast to existing offline learning approaches, we turn the problem on its head: rather than trying to regularise or constrain the policy somehow, we investigate whether the data itself can be enriched using only the available demonstrations and an improved policy derived through a standard BC algorithm, without any additional modifications. To explore this new avenue, we propose a model-based data augmentation method called Trajectory Stitching (TS). Our ultimate aim is to develop a procedure that identifies sub-optimal trajectories and replaces them with better ones. New trajectories are obtained by stitching existing ones together, without the need to generate unseen states. The proposed strategy consists of replaying each existing trajectory in the dataset: for each state-action pair leading to a particular next state along a trajectory, we ask whether a different action could have been taken instead, which would have landed at a different seen state from a different trajectory. An actual jump to the new state only occurs when generating such an action is plausible and it is expected to improve the quality of the original trajectory - in which case we have a *stitching event*.

An illustrative representation of this procedure can be seen in Figure 1, where we assume to have at our disposal only three historical trajectories. In this example, a trajectory has been improved through two stitching events. In practice, to determine the stitching points, TS uses a probabilistic view of state-reachability that depends on learned dynamics models of the environment. These models are evaluated only on in-distribution states enabling accurate prediction. In order to assess the expected future improvement introduced by a potential stitching event, we utilise a state-value function and reward model. Thus, TS can be thought of as a data-driven, automated procedure yielding highly plausible and higher-quality demonstrations to facilitate supervised learning; at the

same time, sub-optimal demonstrations are removed altogether whilst keeping the diverse set of seen states.

Demonstrations can be used to guide RL, to improve on the speed-up of learning of online RL. In these cases, BC can be used to initialise or regularise the training policy [53, 49]. Running TS on the datasets beforehand could be used to improve on the sample efficiency further as the initialised policies will be better; as well as regularising the policy towards an improved one. In future work we aim to leverage TS as a launchpad for online RL. Specifically, an improved BC policy would be useful in improving the sample efficiency for planning [2, 64] as well as deployment efficiency in offline-to-online RL [25, 66].

Our experimental results show that TS produces higher-quality data, with BC-derived policies always superior than those inferred on the original data. Remarkably, we demonstrate that TS-augmented data allow BC to compete with SOTA offline RL algorithms on highly complex continuous control openAI gym tasks implemented in MuJoCo using the D4RL offline benchmarking suite [20]. In terms of a larger system, BC-derived policies are used as a prior to many methods, so a reasoned approach to improving the BC policy could improve these methods also.

## 2 Problem setup

We consider the offline RL problem setting, which consists of finding an optimal decision-making policy from a fixed dataset. The policy is a mapping from states to actions, $\pi : \mathcal{S} \to \mathcal{A}$, whereby $\mathcal{S}$ and $\mathcal{A}$ are the state and action spaces, respectively. The dataset is made up of transitions $\mathcal{D} = \{(s_i, a_i, r_i, s_i')\}$, of current state, $s_i$; action performed in that state, $a_i$; the state in which the action takes the agent, $s_i'$; and the reward for transitioning, $r_i$. The actions have been taken by an unknown behaviour policy, $\pi_\beta$, acting in a Markov decision process (MDP). The MDP is defined as $\mathcal{M} = (\mathcal{S}, \mathcal{A}, \mathcal{P}, \mathcal{R}, \gamma)$, where $\mathcal{P} : \mathcal{S} \times \mathcal{A} \times \mathcal{S} \to [0, 1]$ is the transition probability function which defines the dynamics of the environment, $\mathcal{R} : \mathcal{S} \times \mathcal{A} \times \mathcal{S} \to \mathbb{R}$ is the reward function and $\gamma \in (0, 1]$ is a scalar discount factor [59].

In offline RL, the agent must learn a policy, $\pi^*(a|s)$, that maximises the returns defined as the expected sum of discounted rewards, $\mathbb{E}_\pi[\sum_{t=0}^\infty r_t \gamma^t]$, without ever having access to $\pi_\beta$. Here we are interested in performing imitation learning through BC, which mimics $\pi_\beta$ by performing supervised learning on the state-action pairs in $\mathcal{D}$ [51, 52]. More specifically, assuming a deterministic policy, BC minimises

$$\pi^{\text{BC}}(s) = \arg\min_\pi \mathbb{E}_{s,a\sim\mathcal{D}}[(\pi(s) - a)^2]. \tag{1}$$

The resulting policy also minimises the KL-divergence between the trajectory distributions of the learned policy and $\pi_\beta$ [34]. Our objective for TS is to improve the dataset, by replacing existing trajectories with high-return ones, so that BC can extract a higher-performing behaviour policy than the original. Many offline RL algorithms bias the learned policy towards the behaviour-cloned one [2, 21, 64] to ensure the policy does not deviate too far from the behaviour policy. Being able to extract a high-achieving policy would be useful in many of these offline RL methods.

## 3 Trajectory Stitching

**Overview.** The proposed data augmentation method, Trajectory Stitching, augments $\mathcal{D}$ by stitching together high value regions of different trajectories. Stitching events are discovered by searching for candidate next states which lead to higher returns. These higher quality states are determined by a state-value function, $V(s)$, which is trained using the historical data. This function is unaffected by distributional shift due to only being evaluated on in-distribution states.

Suppose that the transition $(s, a, s')$ came from some trajectory $\mathcal{T}_i$ in $\mathcal{D}$, for which the joint density function is $p(s, a, s') \propto p(s'|s)p(a|s, s')$; here, $p(s'|s)$ represents the environment's forward dynamics and $p(a|s, s')$ is its inverse dynamics. Our aim is to replace $s'$ and $a$ with a candidate next state, $\hat{s}'$ and connecting action $\hat{a}$, which leads to higher returns. To generate a new transition, first we look for a candidate next state, $\hat{s}' \neq s'$, amongst all the states in $\mathcal{D}$, that has been visited by any other trajectory. A suitable criterion to evaluate next state candidates is given by the forward dynamics; conditional on $s$, we require that the new next state must be at least as likely to have been observed as $s'$, i.e. we impose $p(\hat{s}'|s) \geq p(s'|s)$. To be beneficial, the candidate next state must not only be

likely to be reached from $s$ under the environment dynamics, but must also lead to higher returns compared to the current next state. Thus, we also require that, under the pre-trained state-value function, $V(\hat{s}') > V(s')$. Where both these conditions are satisfied, a plausible action connecting $s$ and the newly found $\hat{s}'$ is obtained by finding an action that maximises the inverse dynamics, i.e. $\arg\max_{\hat{a}} p(\hat{a}|s, \hat{s}')$. When the process is completed, we have a *stitching event*.

For each trajectory $\mathcal{T}_i$ in $\mathcal{D}$, we sequentially consider all its transitions $(s, a, s')$ until a stitching event takes place, which leads to a different trajectory, $\mathcal{T}_j$. This process is then repeated for $\mathcal{T}_j$, starting at the current state, until no more stitching events are possible. For example, let us have two trajectories $\mathcal{T}_1$ and $\mathcal{T}_2$, with lengths $N$ and $M$ respectively. TS stitches time point $n$ in $\mathcal{T}_1$ to time point $m$ in $\mathcal{T}_2$ which would lead to a new trajectory to replace $\mathcal{T}_1$,

$$(s_1^{(1)}, a_1^{(1)}, s_2^{(1)}, \ldots, s_{n-1}^{(1)}, a_{n-1}^{(1)}, s_n^{(1)}, \hat{a}, s_m^{(2)}, a_m^{(2)}, s_{m+1}^{(2)}, \ldots, a_{M-1}^{(2)}, s_M^{(2)}).$$

Here $s_j^{(i)}, a_j^{(i)}$ represents a state-action pair for $\mathcal{T}_i$ at time point $j$. Upon completing this process, we have created new and plausible trajectories, under the empirical state distribution, with overall higher expected cumulative returns.

In practice, we do not assume that the forward dynamics, inverse dynamics, reward function and state-value function are known; hence they need to be estimated from the available data. In the remainder of this section we describe the models used to infer these quantities. Algorithm 1 (see Appendix) details the full TS procedure.

**Next state search via a learned dynamics model.** The search for a candidate next state requires a learned forward dynamics model, i.e. $p(s'|s)$. Model-based RL approaches typically use such dynamics' models conditioned on the action as well as the state to make predictions [30, 63, 36, 2]. Here, we use the model differently, only to guide the search process and identify of a suitable next state to transition to. Specifically, conditional on $s$, the dynamics model is used to assess the relative likelihood of observing any other $s'$ in the dataset compared to the observed one.

The environment dynamics are assumed to be Gaussian, and we use a neural network to predict the mean vector and covariance matrix, i.e. $\hat{p}_\xi(s_{t+1}|s_t) = \mathcal{N}(\mu_\xi(s_t), \Sigma_\xi(s_t))$; here, $\xi$ indicate the parameters of the neural network. Modelling the environment dynamics as a Gaussian distribution is common for continuous state-space applications [30, 63, 36, 62]. Furthermore, we take an ensemble $\mathcal{E}$ of $N$ dynamics models, $\{\hat{p}_\xi^i(s_{t+1}|s_t) = \mathcal{N}(\mu_\xi^i, \Sigma_\xi^i)\}_{i=1}^N$. Each model is trained via maximum likelihood estimation so it minimises the following loss

$$\mathcal{L}_{\hat{p}}(\xi) = \mathbb{E}_{s,s'\sim\mathcal{D}}[(\mu_\xi(s) - s')^T \Sigma_\xi^{-1}(s)(\mu_\xi(s) - s') + \log|\Sigma_\xi(s)|],$$

where $|\cdot|$ refers to the determinant of a matrix. Each model's parameter vector is initialised differently; using such an ensemble strategy has been shown to take into account the epistemic uncertainty, i.e. the uncertainty in the model parameters [7, 12, 2, 62].

Once the models have been fitted, to decide whether $\hat{s}'$ can replace $s'$ along any trajectory, we take a conservative approach by requiring that

$$\min_{i\in\mathcal{E}} \hat{p}_\xi^i(\hat{s}'|s) > \operatorname*{mean}_{i\in\mathcal{E}} \hat{p}_\xi^i(s'|s).$$

where the minimum and mean are taken over the ensemble $\mathcal{E}$ of dynamics models.

**Value function estimation and reward prediction model.** Value functions are widely used in reinforcement learning to determine the quality of an agent's current position [59]. In our context, we use a state-value function to assess whether a candidate next state offers a potential improvement over the original next state. To accurately estimate the future returns given the current state, we calculate a state-value function dependent on the behaviour policy of the dataset. The function $V_\theta(s)$ is approximated by a MLP neural network parameterised by $\theta$. The parameters are learned by minimising the squared Bellman error [59],

$$\mathcal{L}_V(\theta) = \mathbb{E}_{s,r,s'\sim\mathcal{D}}[(r + \gamma V_\theta(s') - V_\theta(s))^2]. \tag{2}$$

$V_\theta$ is only used to observe the value of in-distribution states, thus avoiding the OOD issue when evaluating value functions which occurs in offline RL. The value function will only be queried once a candidate new state has been found such that $p(\hat{s}'|s) \geq p(s'|s)$.

Value functions require rewards for training, therefore a reward must be estimated for unseen tuples $(s, \hat{a}, \hat{s}')$. To this end, we train a conditional Wasserstein-GAN [24, 3] consisting of a generator, $G_\phi$ and a discriminator $D_\psi$, with parameters of the neural networks $\phi$ and $\psi$ respectively. A Wasserstein GAN is used due to the training stability over GANs [3], as well as their predictive performance over MLPs and VAEs. The discriminator takes in the state, action, reward, next state and determines whether this transition is from the dataset. The generator loss function is:

$$\mathcal{L}_G(\phi) = \mathbb{E}_{\substack{z \sim p(z) \\ s,a,s' \sim \mathcal{D} \\ \tilde{r} \sim G_\phi(z,s,a,s')}} [D_\psi(s, a, s', \tilde{r})].$$

Here $z \sim p(z)$ is a noise vector sampled independently from $\mathcal{N}(0, 1)$, the standard normal. The discriminator loss function is:

$$\mathcal{L}_D(\psi) = \mathbb{E}_{s,a,r,s' \sim \mathcal{D}}[D_\psi(s, a, s', r)] - \mathbb{E}_{\substack{z \sim p(z) \\ s,a,s' \sim \mathcal{D} \\ \tilde{r} \sim G_\phi(z,s,a,s')}} [D_\psi(s, a, s', \tilde{r})].$$

Once trained, a reward will be predicted for the stitching event when a new action has been generated between two previously disconnected states.

**Action generation via an inverse dynamics model.** Sampling a suitable action that leads from $s$ to the newly found state $\hat{s}'$ requires an inverse dynamics model. Specifically, we require that a synthetic action must maximise the estimated conditional density, $p(a|s, \hat{s}')$. To this end, we train a conditional variational autoencoder (CVAE) [38, 57], consisting of an encoder $q_{\omega_1}$ and a decoder $p_{\omega_2}$ where $\omega_1$ and $\omega_2$ are the respective parameters of the neural networks.

The encoder converts the input data into a lower-dimensional latent representation $z$ whereas the decoder generates data from the latent space. The CVAE objective is to maximise $\log p(a|s, \hat{s}')$ by maximising its lower bound

$$\max_{\omega_1, \omega_2} \log p(a|s, \hat{s}', z) \geq \max_{\omega_1, \omega_2} \mathbb{E}_{z \sim q_{\omega_1}}[\log p_{\omega_2}(a|s, \hat{s}', z)] - D_{\mathrm{KL}}[q_{\omega_1}(z|a, s, \hat{s}')||P(z|s, \hat{s}')],$$

where $z \sim \mathcal{N}(0, 1)$ is the prior for the latent variable $z$, and $D_{\mathrm{KL}}$ represents the KL-divergence [42, 41]. This process ensures that the most plausible action is generated conditional on $s$ and $\hat{s}'$.

**Iterated TS and BC.** TS is run for multiple iterations, updating the value function before each one based on the new data and improved behaviour policy. All other models remain fixed as we do not have any updated information about the underlying MDP. From the new dataset, we extract the behaviour policy using BC, minimising Equation (1). We train BC for 100k gradient steps, reporting the best policy from checkpoints of every 10k steps from 40k onwards. This ensures that BC has trained enough and does not overfit.

## 4 Experimental results

In this section, we provide empirical evidence that TS can produce higher-quality datasets, compared to the original data, by showing BC infers improved policies without collecting any more data from the environment. We call a BC policy run on a TS dataset TS+BC. We compare our method with selected offline RL methods using D4RL datasets. This is to give an insight into how much TS can improve BC by reaching the SOTA performance level of offline RL.

**Performance assessment on D4RL data.** To investigate the benefits of TS+BC as an offline policy learning strategy, we compare its performance with selected state-of-the-art offline RL methods: TD3+BC [21], IQL [40], MBOP [2] and Diffuser [29]. These baselines represent model-free and model-based methods and achieve top results. We make the comparisons on the D4RL [20] benchmarking datasets of the openAI gym MuJoCo tasks; see Table 1. Three complex continuous environments are tested: Hopper, Halfcheetah and Walker2d, with different levels of difficulty. The "medium" datasets were gathered by the original authors using a single policy produced from the early-stopping of an agent trained by soft actor-critic (SAC) [26, 27]. The "medium-replay" datasets are the replay buffers from the training of the "medium" policies. The "expert" datasets were obtained from a policy trained to an expert level, and the "medium-expert" datasets are the combination of both the "medium" and "expert" datasets. In all the cases we have considered, TS+BC outperforms

| Dataset | TD3+BC | IQL | MBOP | Diffuser | BC | TS+BC (ours) |
|---|---|---|---|---|---|---|
| hopper-medium | 59.3 | 66.3 | 48.8 | 58.5 | 55.3 | $64.3 \pm 4.2 (+16.3\%)$ |
| halfcheetah-medium | 48.3 | 47.4 | 44.6 | 44.2 | 42.9 | $43.2 \pm 0.3 (+0.7\%)$ |
| walker2d-medium | 83.7 | 78.3 | 41.0 | 79.7 | 75.6 | $78.8 \pm 1.2 (+4.2\%)$ |
| **Average-medium** | **63.8** | **64.0** | 44.8 | 60.8 | 57.9 | **62.1** |
| hopper-medexp | 98.0 | 91.5 | 55.1 | 107.2 | 62.3 | $94.8 \pm 11.7 (+52.2\%)$ |
| halfcheetah-medexp | 90.7 | 86.7 | 105.9 | 79.8 | 60.7 | $86.9 \pm 2.5 (+43.2\%)$ |
| walker2d-medexp | 110.1 | 109.6 | 70.2 | 108.4 | 108.2 | $108.8 \pm 0.5 (+0.6\%)$ |
| **Average-medexp** | **99.6** | **95.9** | 77.1 | **98.5** | 77.1 | **96.8** |
| hopper-medreplay | 60.9 | 94.7 | 12.4 | 96.8 | 29.6 | $50.2 \pm 17.2 (+69.6\%)$ |
| halfcheetah-medreplay | 44.6 | 44.2 | 42.3 | 42.2 | 38.5 | $39.8 \pm 0.6 (+3.4\%)$ |
| walker2d-medreplay | 81.8 | 73.9 | 9.7 | 61.2 | 34.7 | $61.5 \pm 5.6 (+77.2\%)$ |
| **Average-medreplay** | 62.4 | **70.9** | 21.5 | 66.7 | 34.3 | 50.5 |
| hopper-expert | 107.8 | - | - | - | 111.0 | $111.8 \pm 0.5 (+0.7\%)$ |
| halfcheetah-expert | 96.7 | - | - | - | 92.9 | $93.2 \pm 0.6 (+0.3\%)$ |
| walker2d-expert | 110.2 | - | - | - | 109.0 | $108.9 \pm 0.2 (-0.1\%)$ |
| **Average-expert** | **104.9** | - | - | - | **104.3** | **104.6** |

Table 1: Average normalised scores achieved on three locomotion tasks (Hopper, Halfcheetah and Walker2d) using the D4RL v2 data sets. The results for competing methods have been gathered from the original publications. Bold scores represent values within $5\%$ of the highest average score of the levels of difficulty. TS+BC: In brackets we report the percentage improvement achieved by BC after TS relative to the BC baseline.

the BC baseline, showing that TS creates a higher quality dataset as claimed. Also, while only ever using BC to obtain the final policy, TS+BC is very competitive with current state-of-the-art offline RL methods, especially for the medium, medium-expert and expert datasets. For medium-replay datasets, although TS+BC still attains much higher performing policies than the original BC, we observe lower competitiveness against other offline DRL methods. Due to the way these datasets have been developed, they would appear to be more naturally suited to dynamical programming-based algorithms.

**Implementation details.** Calculating $p(s'|s)$ for all $s' \in \mathcal{D}$ may be computationally inefficient. To speed this up in the MuJoCo environments, we initially select a smaller set of candidate next states by thresholding the Euclidean distance. Although on its own a geometric distance would not be sufficient to identify stitching events, we found that in our environments it can help reduce the set of candidate next states thus alleviating the computational workload.

To pre-select a smaller set of candidate next states, we use two criteria. Firstly, from a transition $(s, a, r, s') \in \mathcal{D}$, a neighbourhood of states around $s$ is taken and the following state in the trajectory is collected. Secondly, all the states in a neighbourhood around $s'$ are collected. This process ensures all candidate next states are geometrically-similar to $s'$ or are preceded by geometrically-similar states. The neighbourhood of a state is an $\epsilon-$ball around the state. When $\epsilon$ is large enough, we can retain all feasible candidate next states for evaluation with the forward dynamic model. Figure 4 (see Appendix) illustrates this procedure.

## 5   Conclusions

In this paper, we have proposed a data augmentation strategy, Trajectory Stitching, which can be applied to historical datasets containing demonstrations of sequential decisions taken to solve a complex task. Without further interactions with the environment, TS can improve the quality of the demonstrations, which in turn has the effect of boosting the performance of BC-extracted policies significantly. This method could be used to extract an improved explicit behavioural cloning policy regulariser for offline RL. This would be specifically important for an offline planning algorithm. TS+BC can be leveraged further for online RL, where the learned policy can be initialised using BC

or the sample efficiency of the algorithm is improved by regularising towards the behavioural policy of demonstrations.

BC is used in many offline RL algorithms, such as a prior policy in offline planning [2, 64] and as a policy regulariser [21]. Although in this paper we have not explored the potential benefits of combining TS with offline reinforcement learning algorithms, our results on the D4RL benchmarking datasets show that TS improves over the initial BC policy, and can in fact reach the level of state-of-the-art offline RL methods. This suggests that many methods could be improved by employing TS either to find a better behavioural cloned policy or by enhancing an initial dataset. We expect TS to be used as a "first- step" to fully leverage the given dataset, enriching the dataset by adding highly-likely (under the environment dynamics) transitions.

Upon acceptance of this paper, we learned about a related methodology called BATS (Best Action Trajectory Stitching) [10]. BATS augments the dataset by adding transitions from planning using a learned model of the environment. Our model-based TS approach differs from BATS in a number of fundamental ways. First, BATS takes a geometric approach to defining state similarity; state-actions are rolled-out using the dynamics model until a state is found that is within $\delta$ of a state in the dataset. Using geometric distances is often inappropriate; e.g. two states may be close in Euclidean distance, yet reaching one from another may be impossible (e.g. in navigation task environments where walls or other obstacles preclude reaching a nearby state). As such, our stitching events are based on the dynamics of the environment and are only assessed between two in-distribution states. Second, BATS allows stitches between states that are $k$-steps apart; this means the reward function needs to be penalised to favour state-action pairs in the original dataset, as model error can compound resulting in unlikely rollouts. In contrast, we only allow one-step stitching between in-distribution states and use the value function to extend the horizon rather than a learned model, this means all our models can be trusted to give accurate predictions without the need for penalties. Finally, BATS adds all stitched actions to the original dataset, then create a new dataset by running value iteration, which is eventually used to learn a policy through BC. This raises many questions about the number of new trajectories need to be collected in this way to extract an optimal policy using BC, as well as other policy learning approaches more suitable to this set up. Our method, is much more suited to policy learning by BC, as after performing TS we are left with a dataset with only high-quality trajectories, where the low-value parts are removed after the stitching event.

We believe that model-based TS opens up a number of directions for future work. For example, it can be extended to multi-agent offline policy learning, for which initial attempts have been made to control the distributional shift with numerous agents [61]. TS could even be used without the value function to increase the heterogeneity of the data without collecting new data. This could be used in conjunction with other offline imitation learning methods [9, 19]. This line of investigation would specifically be useful in situations where collecting new data is expensive or dangerous, but learning from a larger, more heterogeneous data set with additional coverage is expected to improve performance.

# 6   Acknowledgements

CH acknowledges support from the Engineering and Physical Sciences Research Council through the Mathematics of Systems Centre for Doctoral Training at the University of Warwick (EP/S022244/1). GM acknowledges support from a UKRI AI Turing Acceleration Fellowship (EP/V024868/1).

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

# 7 Appendix

## 7.1 Related work

**Imitation learning.** Imitation learning methods aim to emulate a policy from expert demonstrations. DAgger [54] is an online learning approach that iteratively updates a deterministic policy; it addresses the state distributional shift problem of BC through an on-policy method for data collection; similarly to TS, the original dataset is augmented, but this involves on-line interactions. GAIL [28] iteratively updates a generative adversarial network [24] to determine whether a state-action pair can be deemed as expert; a policy is then inferred using a trust region policy optimisation step [56]. TS also uses generative modelling, but this is to create data points likely to have come from the data that connect high-value regions. Whereas imitation learning relies on expert demonstrations, TS creates higher quality datasets from existing, possibly sub-optimal data, to improve off-line policy learning.

**Offline reinforcement learning.** Several model-free offline RL methods deal with distributional shift in two ways: 1) by regularising the policy to stay close to actions given in the dataset [23, 43, 60, 31, 67, 21] or 2) by pessimistically evaluating the Q-value to penalise OOD actions [1, 39, 45]. For instance, BCQ [23] uses a VAE to generate likely actions in order to constrain the policy. The TD3+BC algorithm [21] offers a simplified policy constraint approach; it adds a behavioural cloning regularisation term to the policy update biasing actions towards those in the dataset. Alternatively, CQL [45] adjusts the value of the state-action pairs to "push down" on OOD actions and "push up" on in-distribution actions. IQL [40] avoids querying OOD actions altogether by manipulating the Q-value to have a state-value function in the SARSA-style update. All the above methods try to directly deal with OOD actions, either by avoiding them or safely handling them in either the policy improvement or evaluation step. In contrast, TS generates unseen actions between in-distribution states; by doing so, we avoid distributional shift by evaluating a state-value function only on seen states.

Model-based algorithms rely on an approximation of the environment's dynamics [58, 30]. In the online setting, they tend to improve sample efficiency [33, 30, 15, 7, 12]. In an offline learning context, the learned dynamics have been exploited in various ways. For instance, Model-based Offline policy Optimization (MOPO) [63] augments the dataset by performing rollouts using a learned, uncertainty-penalised, MDP. Unlike MOPO, TS does not introduce imagined states, but only actions between reachable unconnected states. Diffuser [29] uses a diffusion probabilistic model to predict a whole trajectory rather than a single state-action pair; it can generate unseen trajectories that have high likelihood under the data and maximise the cumulative rewards of a trajectory ensuring long-horizon accuracy. In contrast, our generative models are not used for planning hence we do not require sampling a full trajectory; instead, our models are designed to only be evaluated locally ensuring one-step accuracy between $s$ and $\hat{s}'$.

**State similarity metrics.** A central aspect of the proposed data augmentation method consists of defining the stitching event, which uses a notion of state similarity to determine whether two states are "close" together. Using geometric distances only would often be inappropriate; e.g. two states may be close in Euclidean distance, yet reaching one from another may be impossible (e.g. in navigation task environments where walls or other obstacles preclude reaching a nearby state). Bisimulation metrics [16] capture state similarity based on the dynamics of the environment. These have been used in RL mainly for system state aggregation [17, 35, 65]; they are expensive to compute [11] and usually require full-state enumeration [4, 5, 14]. A scalable approach for state-similarity has recently been introduced by using a pseudometric [8] which made calculating state-similarity possible for offline RL. PLOFF [14] is an offline RL algorithm that uses a state-action pseudometric to bias the policy evaluation and improvement steps keeping the policy close to the dataset. PLOFF uses a pseudometric to stay close to the data, we can bypass this notion altogether by requiring reachability in one step.

## 7.2 Trajectory Stitching

The full procedure for the Trajectory stitching method is outlined in Algorithm 1.

---
**Algorithm 1** Trajectory Stitching
---
**Initialise:** An action generator $p_{\omega_1}$, a reward generator $G_\phi$, an ensemble of dynamics models $\{\hat{p}^i_\xi(s'|s)\}^N_{i=1}$, an acceptance threshold $\tilde{p}$, and a dataset $\mathcal{D}_0$ made up of $T$ trajectories $(\mathcal{T}_1, \ldots \mathcal{T}_T)$

1: **for** $k = 0, \ldots, K$ **do**
2:     Train state-value function, $V$ on $\mathcal{D}_k$ by minimising Equation (2).
3:     **for** $t = 1, \ldots, T$ **do**
4:         Select $s, s' = s_0, s'_0 \in \mathcal{T}_t$
5:         Initialise new trajectory, $\hat{\mathcal{T}}_t$
6:         **while** not done **do**
7:             Create set of candidate states from neighbourhood, $\{\hat{s}'_j\}^N_{j=1} \sim$ Neighbourhood
8:             Evaluate dynamics models for new set of states and take minimum, $\min_i \hat{p}^i_{\xi,\pi}(\hat{s}'|s)$
9:             **if** $\min_i \hat{p}^i_\xi(\hat{s}'_j|s) > \text{mean}_i \hat{P}^i_\xi(s'|s)$, $V(\hat{s}'_j) = \max_i V(\hat{s}'_i)$ and $V(\hat{s}'_j) > V(s')$ **then**
10:                Generate a new action and reward,
                     $\tilde{a} \sim p_{\omega_1}(z, s, \hat{s}'_j), \quad \tilde{r} \sim G_\phi(z, s, \tilde{a}, \hat{s}'_j)$
11:                Add $(s, \tilde{a}, \tilde{r}, \hat{s}'_j)$ to new trajectory $\hat{\mathcal{T}}_t$
12:                Set $s = \hat{s}'_j$
13:             **else**
14:                Add original transition, $(s, a, r, s')$ to the new trajectory $\hat{\mathcal{T}}_t$
15:                Set $s = s'$
16:             **end if**
17:         **end while**
18:         **if** $\sum_{\hat{\mathcal{T}}_t} r_i > (1 + \tilde{p}) * \sum_{\mathcal{T}_t} r_i$ **then**
19:             $\hat{\mathcal{T}}_t = \hat{\mathcal{T}}_t$
20:         **else**
21:             $\hat{\mathcal{T}}_t = \mathcal{T}_t$
22:         **end if**
23:     **end for**
24:     Collect trajectories into dataset, $\mathcal{D}_{k+1} = (\hat{\mathcal{T}}_1, \ldots \hat{\mathcal{T}}_T)$
25: **end for**
---

## 7.3 Further Experiments

**Expected performance on sub-optimal data.** BC minimises the KL-divergence of trajectory distributions between the learned policy and $\pi_\beta$ [34]. As TS has the effect of improving $\pi_\beta$, this suggests that the KL-divergence between the trajectory distributions of the learned policy and the expert policy would be smaller post TS. To investigate this, we used two complex locomotion tasks, Hopper and Walker2D, in OpenAI's gym [6]. Independently for each task, we first train an expert policy, $\pi^*$, with TD3 [22], and use this policy to generate a baseline noisy dataset by sampling the expert policy in the environment and adding white noise to the actions, i.e. $a = \pi^*(s) + \epsilon$. A range of different, sub-optimal datasets are created by adding a certain amount of expert trajectories to the noisy dataset so that they make up $x\%$ of the total trajectories. Using this procedure, we create eight different datasets by controlling $x$, which took values in the set $\{0, 0.1, 2.5, 5, 10, 20, 30, 40\}$. BC is run on each dataset for 5 random seeds. In all experiments we run TS for five iterations, as this provides enough to increase the quality of the data without being overly computationally expensive (see the Appendix for results across different iterations). We run TS (for five iterations) on each dataset over three different random seeds and then create BC policies over the 5 random seeds, giving 15 TS+BC policies. Random seeds, cause different TS trajectories as they affect the latent variables sampled for the reward function and inverse dynamics model. Also, the initialisation of weights is randomised for the value function and BC policies, so the robustness of the methods is tested over multiple seeds.

Figure 3 shows the scores as average returns from 10 trajectory evaluations of the learned policies. TS+BC consistently improves on BC across all levels of expertise, for both the Hopper and Walker2d environments. As the percentage of expert data increases, TS is available to leverage more high-value transitions, consistently improving over the BC baseline. Figure 2 (left) shows the average difference in KL-divergences of the BC and TS+BC policies against the expert policy. Precisely, the

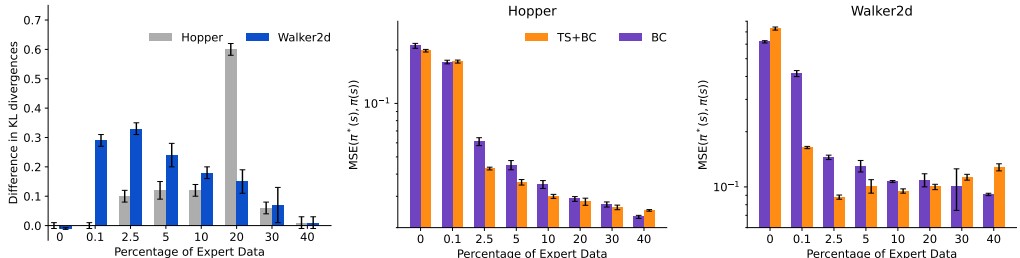

Figure 2: Estimated KL-divergence and MSE of the BC and TS+BC policies on the Hopper and Walker2d environments as the fraction of expert trajectories increases. (Left) Relative difference between the KL-divergence of the BC policy and the expert and the KL-divergence of the TS+BC policy and the expert. Larger values represent the TS+BC policy being closer to the expert than the BC policy. MSE between actions evaluated from the expert policy and the learned policy on states from the Hopper (Middle) and Walker2d (Right) environments. The y-axes (Middle and Right) are on a log-scale. All policies were collected by training BC over 5 random seeds, with TS being evaluated over 3 different random seeds. All KL-divergences were scaled between 0 and 1, depending on the minimum and maximum values per task, before the difference was taken.

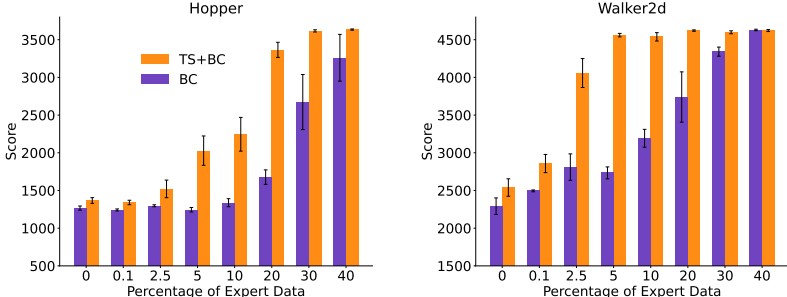

Figure 3: Comparative performance of BC and TS+BC as the fraction of expert trajectories increases up to $40\%$. For two environments, Hopper (left) and Walked2D (right), we report the average return of 10 trajectory evaluations of the best checkpoint during BC training. BC has been trained over 5 random seeds and TS has produced 3 datasets over different random seeds.

y-axis represents $D_{KL}(\rho_{\pi^*}(\tau), \rho_{\pi^{BC}}(\tau)) - D_{KL}(\rho_{\pi^*}(\tau), \rho_{\pi^{TS+BC}}(\tau))$, where $\rho_\pi(\tau)$ is the trajectory distribution for policy $\pi$. So, a positive value represents the TS+BC policy being closer to the expert, and a negative value represents the BC policy being closer to the expert, with the absolute value representing the degree to which this is the case. We also scale the average KL-divergence between $0$ and $1$, where $0$ is the smallest KL-divergence and $1$ is the largest KL-divergence, per task. This makes the scale comparable between Hopper and Walker2d. The KL divergences are calculated following [34], $D_{KL}(\rho_{\pi^*}(\tau), \rho_\pi(\tau)) = \mathbb{E}_{s\sim\rho_{\pi^*}, a\sim\pi^*(s)}[\log \pi^*(a|s) - \log \pi(a|s)]$. The Figure shows that BC can extract a behaviour policy closer to the expert after performing TS on the dataset, except in the $0\%$ case for Walker2D, however the difference is not significant. TS seems to work particularly well with a minimum of $2.5\%$ expert data for Hopper and $0.1\%$ for Walker2d.

Furthermore, Figure 2 (middle and right) shows the mean square error (MSE) between actions from the expert policy and the learned policy for the Hopper (middle) and Walker2d (right) tasks. Actions are selected by collecting 10 trajectory evaluations of an expert policy. As we expect, the TS+BC policies produce actions closer to the experts on most levels of dataset expertise. The only surprising result is that for $0\%$ expert data on the Walker2d environment the BC policy produces actions closer to the expert than the TS+BC policy. This is likely due to TS not having any expert data to leverage and so cannot produce any expert trajectories. However, TS still produces a higher-quality dataset than previous as shown by the increased performance on the average returns. This offers empirical confirmation that TS does have the effect of improving the underlying behaviour policy of the dataset.

### 7.4 Further implementation details

In this section we report on all the hyperparameters required for TS as used on the D4RL datasets. All hyperparameters have been kept the same for every dataset, notable the acceptance threshold of $\tilde{p} = 0.1$. TS consists of four components: a forward dynamics model, an inverse dynamics model, a reward function and a value function. Table 2 provides an overview of the implementation details and hyperparameters for each TS component. As our default optimiser we have used Adam [37] with default hyperparameters, unless stated otherwise.

**Forward dynamics model.** Each forward dynamics model in the ensamble consists of a neural network with three hidden layers of size 200 with ReLU activation. The network takes a state $s$ as input and outputs a mean $\mu$ and standard deviation $\sigma$ of a Gaussian distribution $\mathcal{N}(\mu, \sigma^2)$. For all experiments, an ensemble size of 7 is used with the best 5 being chosen.

**Inverse dynamics model.** To sample actions from the inverse dynamics model of the environment, we have implemented a CVAE with two hidden layers with ReLU activation. The size of the hidden layer depends on the size of the dataset [67]: when the dataset has less than $900,000$ transitions (e.g. the medium-replay datasets) the layer has 256 nodes; when larger, it has 750 nodes. The encoder $q_{\omega_1}$ takes in a tuple consisting of state, action and next state; it encodes it into a mean $\mu_q$ and standard deviation $\sigma_q$ of a Gaussian distribution $\mathcal{N}(\mu_q, \sigma_q)$. The latent variable $z$ is then sampled from this distribution and used as input for the decoder along with the state, $s$, and next state, $s'$. The decoder outputs an action that is likely to connect $s$ and $s'$. The CVAE is trained for $400,000$ gradient steps with hyperparameters given in Table 2.

**Reward function.** The reward function is used to predict reward signals associated with new transitions, $s, a, s'$. For this model, we use a conditional-WGAN with two hidden layers of size 512. The generator, $G_\phi$, takes in a state $s$, action $a$, next state $s'$ and latent variable $z$; it outputs a reward $r$ for that that transition. The decoder takes a full transition of $(s, a, r, s')$ as input to determine whether this transition is likely to have come from the dataset or not.

**Value function.** Similarly to previous methods [23], our value function $V_\theta$ takes the minimum of two value functions, $\{V_{\theta_1}, V_{\theta_2}\}$. Each value function is a neural network with two hidden layers of size 256 and a ReLU activation. The value function takes in a state $s$ and determines the sum of future rewards of being in that state and following the policy (of the dataset) thereon.

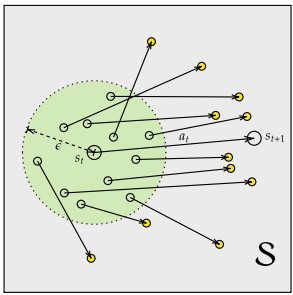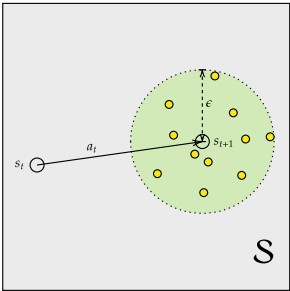

Figure 4: Visualisation of our two definitions of a neighbourhood. For a transition $(s_t, a_t, s_{t+1}) \in \mathcal{D}$, the neighbourhoods are used to reduce the size of the set of candidate next states. (Left) All states within an $\epsilon$-ball of the current state, $s_t$, are taken and the next state in their respective trajectories (joined by an action shown as an arrow) are added to the set of candidate next states. (Right) All states within an $\epsilon$-ball of the next state, $s_{t+1}$ are added to the set of candidate next states. The full set of candidate next states are highlighted in yellow.

**KL-divergence experiment.** As the KL-divergence requires a continuous policy, the BC policy network is a 2-layer MLP of size 256 with ReLU activation, but with the final layer outputting the parameters of a Gaussian, $\mu_s$ and $\sigma_s$. We carry out maximum likelihood estimation using a batch size of 256. For the Walker2d experiments, TS was slightly adapted to only accept new trajectories

| | Hyperparameter | Value |
|---|---|---|
| Forward Dynamics model | Optimiser | Adam |
| | Learning rate | 3e-4 |
| | Batch size | 256 |
| | Ensemble size | 7 |
| Inverse Dynamics model | Optimiser | Adam |
| | Learning rate | 1e-4 |
| | Batch size | 100 |
| | Latent dim | 2*action dim |
| Reward Function | Optimiser | Adam $\beta = (0.5, 0.999)$ |
| | Learning rate | 1e-4 |
| | Batch size | 256 |
| | Latent dim | 2 |
| | L2 regularisation | 1e-4 |
| Value Function | Optimiser | Adam |
| | Learning rate | 3e-4 |
| | Batch size | 256 |

Table 2: Hyperparameters and values for models used in TS

if they made less than ten changes. For each level of difficulty, TS is run 3 times and the scores are the average of the mean returns over 10 evaluation trajectories of 5 random seeds of BC. To compute the KL-divergence, a continuous expert policy is also required, but TD3 gives a deterministic one. To overcome this, a continuous expert policy is created by assuming a state-dependent normal distribution centred around $\pi^*(s)$ with a standard deviation of $0.01$.

**D4RL experiments.** For the D4RL experiments, we run TS 3 times for each dataset and average the mean returns over 10 evaluation trajectories of 5 random seeds of BC, to attain the results for TS+BC. For the BC results, we average the mean returns over 10 evaluation trajectories of 5 random seeds. The BC policy network is a 2-layer MLP of size 256 with ReLU activation, the final layer has $\tanh$ activation multiplied by the action dimension. We use the Adam optimiser with a learning rate of $1e-3$ and a batch size of $256$.

### 7.5 Number of iterations of TS

TS can be repeated multiple times, each time using a newly estimated value function to take into account the newly generated transitions. In all our experiments, we choose 5 iterations. Figure 5 shows the scores of the D4RL environments on the different iterations, with the standard deviation across seeds shown as the error bar. With iteration 0 we indicate the BC score as obtained on the original D4RL datasets. For all datasets, we observe that the average scores of BC increase initially over a few iterations, then remain stable with only some minor random fluctuations. For Hopper and Walker2d medium-replay, there is a higher degree of standard deviation across the seeds, which gives a less stable average as the number of iterations increases.

### 7.6 Ablation study

TS uses a value function to estimate the future returns from any given state. Therefore TS+BC has a natural advantage over just BC which uses only the states and actions. To ensure that using a value function is only sufficient to improve the performance of BC, we have test a weighted version of the BC loss function whereby the weights are given by the estimated value function, i.e.

$$\pi^{\text{BC}}(s) = \arg \min_{\pi} \mathbb{E}_{s,a \sim \mathcal{D}}[V_\theta(s)(\pi(s) - a)^2].$$

This weighted-BC method gives larger weight to the high-value states and lower weight to the low-value states during training. On the Hopper medium and medium-expert datasets, training

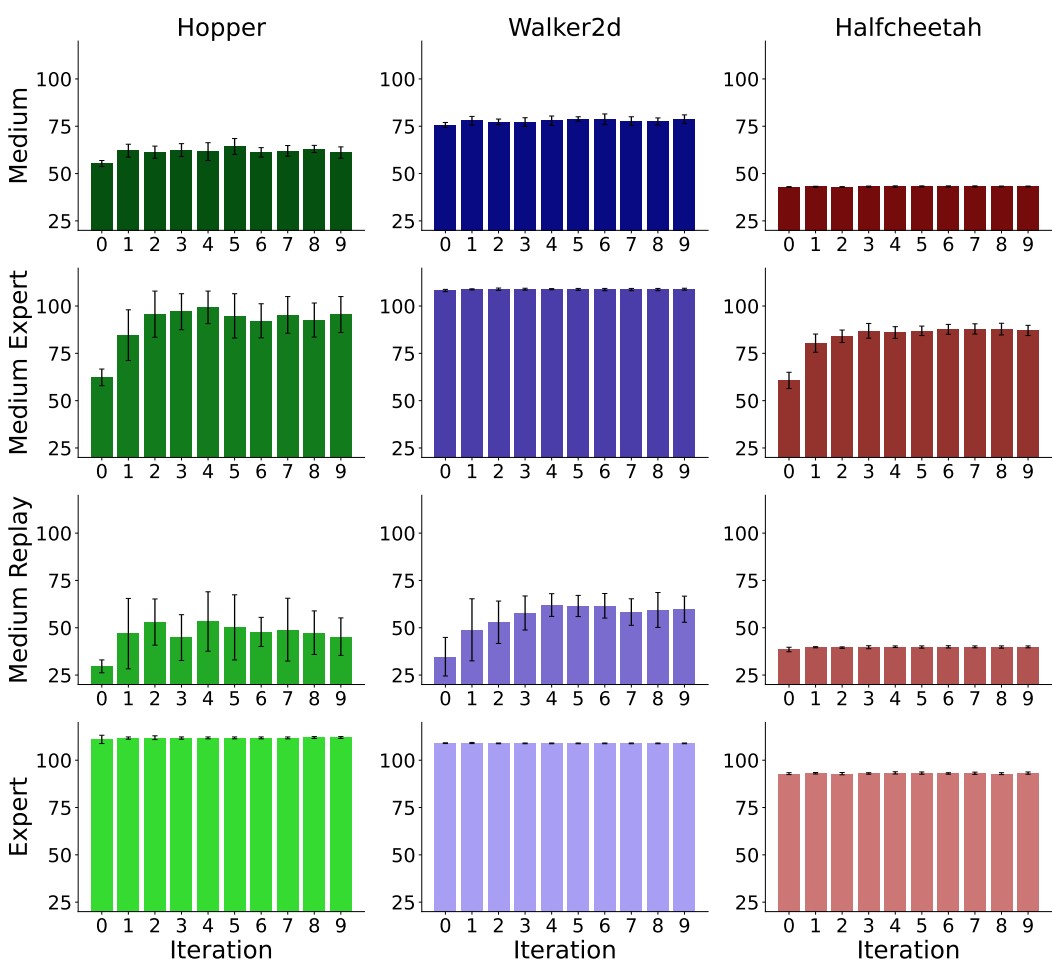

Figure 5: Returns of BC extracted policies as the number of iterations of TS is increased. Iteration 0 are the BC scores on the original D4RL datasets. The errors bars represent the standard deviation of the average returns of 10 trajectory evaluations over 5 random seeds of BC and 3 random seeds of TS.

this weighted-BC method only gives a slight improvement over the original BC-cloned policy. For Hopper-medium, weighted-BC achieves an average score of 59.21 (with standard deviation 3.4); this is an improvement over BC (55.3), but lower lower than TS+BC (64.3). Weighted-BC on hopper-medexp achieves an average score of 66.02 (with standard deviation 6.9); again, this is a slight improvement over BC (62.3), but significantly lower than TS+BC (94.8). The experiments indicate that using a value function to weight the relative importance of seen states when optimising the BC objective function is not sufficient to achieve the performance gains introduced by TS.

