# OpenReview forum: "Model-based Trajectory Stitching for Improved Offline Reinforcement Learning"
_NeurIPS.cc/2022/Workshop/Offline_RL — Offline RL Workshop NeurIPS 2022_

### Official Review · Reviewer_rvqG · 2022-10-18
**Decent workshop paper that could use some clarifying of key contributions.**

**Rating:** 6
**Confidence:** 3

**Review:**

This paper proposes a novel method for improving behavior cloning by allowing trajectories to be modified with a “sticking” operation. The method is novel and interesting, but the evaluation and writing style could be improved.

## strengths
- the method is well motivated in general. Being able to use information from multiple trajectories with an offline stitching process will be very useful in behavior cloning, but also in related fields like model-based reinforcement learning.
- the paper is in general easy to follow and well written. Figure 1 is simple and illustrates the idea well.

## weaknesses
- the method itself is somewhat hard to evaluate in only the offline RL setting. This can be seen by the results looking “okay”. To me, it should be enough to get it accepted in a workshop being novel, but I would encourage the authors to try their trajectory-stitching algorithm in more domains (such as model-based RL / planning)
- I think the paper reads as if the expert only data would showcase their method, but it performs really similarly to vanilla BC in that domain. Also, what would happen if the authors tried TD3+BC+TS?
- the system presented is very complex. To get this accepted into a full conference would take a moderate amount of work analyzing the components and verifying their performance.

## questions
- why don't you have trajectory stitching in the title! The title doesn't really lead me to understand what the paper will talk about.
- could the overview of BC and related methods in the intro be shortened to include more experiments? The intro being so verbose distracts from the methodology and contributions.
- I was wondering if the authors every considered a variant of backward stitching. It doesn’t seem immediately feasible in the same way, but would be interesting to forecast backwards from a goal (and may be useful more outside of BC)
- How was the generative model for states to train the value function evaluated? This seems like it could be very useful in RL. Or, are citations missing from this as being a common practice? I think it could be more of a contribution on its own if it is new.

## nits / other comments
- Seems like this should be cited? https://arxiv.org/abs/2204.12026? How is this similarly named paper related?
- some of the appendix content is really interesting. Supports the case rounding towards accepting this paper to the workshop.

---

### Official Review · Reviewer_FuQf · 2022-10-19

**Rating:** 5
**Confidence:** 5

**Review:**

This paper presents a data augmentation scheme for BC in the offline setting. The authors propose to produce trajectory stitching based on the likelihood of the next state, next action, and high values. The authors train a forward dynamics model, an inverse dynamics model, a value function with a reward generator and a policy. The results show that trajectory stitching + BC can improve BC on D4RL tasks.

Pros:
1. The idea of conducting trajectory stitching for BC is neat and original since BC usually suffers from suboptimal data where offline RL can excel. This idea can be an effective approach to bridging that gap.
2. The empirical result on D4RL clearly shows that the method can improve over BC.

Cons:
1. The method is pretty complex with many components as listed in the summary and there are many hyperparameters/models to tune. It is unclear if such a scheme is general enough in all kinds of domains. I suspect that making it work in various settings could be challenging.
2. It is unclear if the approach can outperform simple baselines such as percentage BC where we simply train on trajectories with top K% return and returned-weighted BC e.g. RWR and AWR.
3. The method is still much worse than offline RL in medium-replay datasets, suggesting that it's not doing a good job of stitching in diverse datasets.